# Current Therapeutic Approaches in Cervical Cancer Based on the Stage of the Disease: Is There Room for Improvement?

**DOI:** 10.3390/medicina59071229

**Published:** 2023-06-30

**Authors:** Irinel-Gabriel Dicu-Andreescu, Augustin-Marian Marincaș, Victor-Gabriel Ungureanu, Sînziana-Octavia Ionescu, Virgiliu-Mihail Prunoiu, Eugen Brătucu, Laurențiu Simion

**Affiliations:** 1Clinical Department No 10, General Surgery, University of Medicine and Pharmacy “Carol Davila”, 050474 Bucharest, Romania; andreescugabriel43@gmail.com (I.-G.D.-A.);; 2Department of Oncological Surgery, Oncological Institute “Alexandru Trestioreanu”, 022328 Bucharest, Romania

**Keywords:** cervical cancer, FIGO, surgery, radiotherapy, oncology, lymph nodes, hysterectomy

## Abstract

Cervical cancer continues to be among the most common malignancies in women, and in recent decades, important measures have been taken to reduce its incidence. The first and most important steps to achieve this goal are oriented toward prevention through screening programs and vaccination, mainly against oncogenic human papillomavirus (HPV) strains 16 and 18. The therapeutic approach is based on the diagnosis and treatment guidelines for cervical cancer, which establish for each stage (FIGO, TNM) specific conduct. These guidelines summarize quite precisely the elements of therapeutic practice, but, in some places, they leave optional variants based on which nuanced approaches could be established. Adherence to these guidelines, which include the performing of minor or major surgery, with or without chemotherapy and radiation therapy, combined with advanced imaging investigations, has been able to lead to a substantial increase in survival. The purpose of this literature review is to discuss the diagnosis and treatment options in cervical cancer depending on the histological type, FIGO staging, and patient performance index, taking into account the hospital resources available in middle-income countries (percentage of gross domestic product allocated to health services around 5.5%, in the case of Romania).

## 1. Introduction

Cervical cancer is the fourth most common cancer in women and the seventh worldwide, with over 600.000 new cases in 2020, 50% of which are diagnosed in advanced stages [1]. In Romania, according to Globocan 2020, the incidence is between 18.6–29.3/100.000—which is double compared to Central and Eastern Europe (incidence 14.5/100.000), and with mortality between 11.4–19/100.000, which is triple compared to the same European region (6.1/100.000) [1]. Moreover, up to a quarter of females are diagnosed under the age of 40 when the psychosocial and sexual impact is the greatest [2,3].

The most well-known risk factor for cervical cancer is infection with the oncogenic strains of human papillomavirus (HPV) 16 and 18 [4]. The most common way of transmission is through sexual intercourse, which is why risk factors include early onset of sexual life, increased number of sexual partners, and immunosuppression [2]. Another major risk factor, although less known than HPV infection, is smoking [5]. The European Prospective Investigation into Cancer and Nutrition (EPIC) study demonstrated that smoking could independently increase the risk of cervical neoplasia two-fold, and its cessation can also decrease it by two-fold [5]. Moreover, passive smoking is also incriminated [6,7].

Body mass index (BMI) is also a possible risk factor. In 2009, in a systematic review with meta-analysis, Maruthur et al. stated that the risk of cervical neoplasia is increased in patients with increased BMI due to the fact that they reported much lower screening compared to patients with normal BMI [8]. More recently, in 2019, Poorolajal et al. found in a meta-analysis that there is only a weak association between BMI over 30 kg/m^2^ and cervical neoplasia and no association with a BMI between 25–30 kg/m^2^ [9]. However, because of all the metabolic disturbances that exist in obesity, patients should be advised to eliminate this potential risk factor, even if not fully confirmed, and to increase their intake of fiber, vegetables, and fruits, a small study claiming a risk reduction with such a diet [10].

Finally, exposure in utero to synthetic estrogens, the most incriminated being diethyl-stilbestrol, can cause grade 2 or higher cervical neoplasia in these women, the cumulative risk being 6.9% vs. 3.4% in those not exposed, a risk that remains raised throughout entire life [11].

The purpose of this review is to explore treatment options depending on the histological type and stage of the disease in an attempt to improve the survival of patients with cervical cancer. The discussions also take into account the particularities of treatment in the countries of Central and Eastern Europe, especially in the case of stages considered loco-regionally advanced, in which surgical treatment can be performed in selected cases.

## 2. Cervical Cancer Screening

Screening for cervical cancer consists of testing for HPV infection or the morphological microscopic examination of the Babes-Papanicolaou smear [12]. These two tests can be associated with each other. They are performed on asymptomatic women after the age of 21 to prevent the appearance of dysplastic or neoplastic lesions and to establish an optimal treatment course depending on the result obtained [12].

Screening tests usually start at the age of 21 and are repeated once every 3 years for the Babes-Papanicolaou test or every 5 years for the HPV test [13]. Screening can stop after the age of 65 if the patient has had the last tests negative, has no personal history of cervical dysplasia, or has a history of surgical interventions that involved excision of the cervix, such as hysterectomy for benign lesions such as uterine fibroadenomas [13].

## 3. History of the Treatment for Cervical Neoplasia

Surgical treatment for cervical neoplasia began to be used at the beginning of the 17th century in patients with uterine prolapse, in which the cervical tumor was visible. However, due to poor results and postoperative complications, local excision of the tumor alone was considered insufficient and, therefore, an intervention without benefit. Only two centuries later, a more extensive procedure was introduced, a simple total hysterectomy, with a transabdominal (Freund) or transvaginal (Czerny) approach and with slightly better survival results [14]. In 1895 Clark presented a study with 12 patients in which he performed wide resection of the parameters along with resection of the uterus, and for some of them, pelvic lymphadenectomy was also performed [15].

In 1898, Ernst Wertheim included, in addition to the simple total hysterectomy and resection of the parameters, also the resection of a larger portion of the vaginal border and excision of the pelvic lymph nodes, a procedure that still bears the author’s name today and is considered the standard radical hysterectomy [16].

In 1902 Thoma Ionescu, the founder of the Romanian School of Surgery and Topographical Anatomy, described in his work “Surgical treatment of uterine cancer” the technique of total radical hysterectomy with pelvic and lumbo-aortic lymphadenectomy [17]. The next steps in the development of the technique were made by Latzko (1919) with a much more anatomical approach, emphasizing the identification of the uterosacral, cardinal, and paracervical ligaments [18].

In Japan, in 1921, Okabayashi considered the Wertheim operation not radical enough, and, in addition to the dissection suggested by Latzko, he introduced the separation of the paracervical ligaments to obtain a longer vaginal resection [19]. Thirty years later, in the United States of America, Meigs also modified the Wertheim technique, adding en-bloc pelvic lymphadenectomy and thereby obtaining some remarkable survival results: 90% at 5 years in stage I and 60% at 5 years in stage II [20].

Finally, Hockel et al. reported in 2003 the so-called radical hysterectomy with mesometrial resection, which broadly maintains the steps of the Wertheim procedure but performs a more extensive pelvic lymphadenectomy [21]. In the same year, Palfalvi and Ungar described the laterally extended parametrectomy with the purpose of radical resection in advanced cancers with positive nodes at the anatomic-pathological examination. This technique is named super-radical hysterectomy, through which the excision of the base of the cardinal ligament is performed along with the internal iliac vessels [22]. Figure 1 describes a schematic history that led to the development of surgical techniques in cervical cancer.

As for other treatment methods, radiotherapy was first used by American surgeon Robert Abbe for cervical cancer in 1910 [23]. Two years later, Forsell used radium in inoperable patients and also obtained cases of total clinical remission [23]. On the other hand, some studies state that the use of radiotherapy per primam is associated with lower survival and subsequent risks of local recurrence or distant metastases [24]. However, at the present moment, according to the National Comprehensive Cancer Network (NCCN) guidelines, radiotherapy in both forms, either external or internal (brachytherapy), is used in almost all stages of cervical neoplasia, either as a singular treatment or as an adjuvant or neo-adjuvant treatment [25].

## 4. Multimodal Treatment of Cervical Cancer According to Disease Stage

The most important factor in the choice of therapeutic conduct is the stage of the disease. Other factors to be taken into account are the histological type—squamous cell carcinoma (the most common) or adenocarcinoma, age, comorbidities, and if possible, fertility preservation [25].

Most commonly, cervical cancer is staged using the International Federation of Gynecology and Obstetrics (FIGO) criteria described in Table 1.

### 4.1. Stage IA1

The depth of stromal invasion is up to 3 mm and can only be diagnosed microscopically. Treatment at this stage depends on the patient’s choice, the maintenance of fertility, and the lymphovascular invasion identified on the biopsy.

If the patient chooses to preserve fertility, cervical conization is recommended [26]. Kim et al. state that this technique has a protective role while also being an independent positive prognostic factor. Therefore, even if later it is decided to perform a hysterectomy, the patients have a higher disease-free survival rate, and the need for adjuvant radiochemotherapy treatment is less common [26]. If the resection margins are negative, conization may represent the definitive treatment [27]. According to the ESGO/ESTRO/ESP guidelines, if the margins are positive, it can be opted for repeating the conization, radical surgical interventions being viewed as overtreatment in this stage [28]. However, the NCCN guidelines maintain radical trachelectomy as a surgical option [2,25,29]. If the lymphovascular invasion is detected, conization with sentinel lymph node biopsy without additional pelvic lymph node dissection is recommended [28]. Additionally, something worth emphasizing is that, at this stage, there are no differences in survival regarding the histological type of neoplasia or the surgical technique chosen, even if one of the main risk factors for recurrence described in the literature is adenocarcinoma [30,31].

If the patient does not want to preserve her fertility, a simple total hysterectomy is chosen, meaning the recurrence rate of the disease is even lower if conization was previously performed [26]. If there is lymphovascular invasion, the recommended intervention is still standard hysterectomy with sentinel lymph node biopsy, again, without additional pelvic lymph node dissection [28].

Pelvic external radiotherapy, followed by brachytherapy, may be an option only if the patient refuses surgery, although the results are controversial. In a study by Yang et al., radiotherapy in stages I and II of cervical cancer resulted in shorter overall and disease-free survival, especially if the neoplasia was diagnosed before menopause [32]. Similarly, when the effect of radiotherapy was analyzed in relation to tumor size, in the case of tumors with dimensions less than 2 cm, the effect was unfavorable [32]. An explanation of this behavior can be the fact that small, “early stage” tumors have an increased ability to defend against the errors in DNA induced by radiotherapy [33].

The prognosis of patients is extremely good, with a 5-year survival of 95.8% [34].

### 4.2. Stage IA2

The depth of stromal invasion is 3–5 mm and does not exceed 7 mm in surface extension [35]. The tumor is diagnosed by microscopy.

The standard surgical treatment is radical hysterectomy with pelvic lymphadenectomy (Wertheim intervention) or radiotherapy [36]. However, to avoid the complications of more invasive methods, simple hysterectomy, trachelectomy, or conization, with or without pelvic lymphadenectomy, can also be considered without significant differences in survival [37].

Radical hysterectomy involves the resection of the cervix, the uterus, the upper part of the vagina, and the parameters [36]. It can be performed classically by laparotomy, as presented in Figure 2, or by laparoscopy [38].

Regarding the laparoscopic approach, a significant debate and controversy in the medical community was produced by the LACC trial, published in 2018, that aimed to compare the efficacy of laparoscopic radical hysterectomy versus abdominal radical hysterectomy in the treatment of early-stage cervical cancer. The initial results suggested a higher rate of disease recurrence and worse survival outcomes in patients who underwent laparoscopic surgery. However, subsequent analysis and discussions raised concerns about potential biases and limitations in the trial design. Therefore, the choice between laparoscopic or open surgery should be made, taking into consideration the stage of the disease, the patient’s option, and the skills of the surgeon [39].

It should be noted that conization is recommended before laparoscopic hysterectomy to minimize the recurrence of the disease [26]. It is considered that a reason for recurrence in patients in whom conization was not performed is the use of the uterus manipulator and repeated traumatization of the tumor during the laparoscopic intervention [26].

Trachelectomy is especially recommended for patients who want to preserve their fertility, and it is also the most used method in this situation. The main risks of this procedure are premature birth and pregnancy loss, which are more frequent than in the general population, especially in the 2nd trimester [29,40]. The approach can be transvaginal or abdominal—the last technique is easier due to its similarity with radical hysterectomy. Additionally, it can be performed minimally invasive—laparoscopic or robotically assisted, in which postoperative recovery is much faster. The intervention is completed by lymph node assessment performed either by imaging explorations or by the sentinel node technique and pelvic lymphadenectomy [29]. The sentinel node technique, often used for breast cancer or malignant melanoma, has the potential to limit the complications of lymphadenectomy [41].

The prognosis of patients is extremely good, with a 5-year survival of 95% [34].

### 4.3. Stages IB1, IB2, IIA1

Tumors invade the stroma to a depth of at least 5 mm, with lesions confined to the cervix and surface extension up to 4 cm [35].

If the patient wishes to preserve her fertility, radical trachelectomy with extended pelvic or even para-aortic lymphadenectomy can be performed [25]. If the tumor is larger than 2 cm but less than 4 cm (stage IB2), neoadjuvant therapy can be an option. It is represented either by preoperative brachytherapy, which can minimize the indication of adjuvant radiotherapy, or by chemotherapy, generally based on a combination of platinum salts and paclitaxel. Chemotherapy is not established as a standard treatment at this stage because there are still not enough studies to demonstrate its efficiency. However, combined with brachytherapy may subsequently allow a limited surgical intervention with minimal impact on fertility [40,42]. This is possible up to stage IB2 and is associated with a disease recurrence rate of 10% and a mortality rate of 2.9% [40].

If the patient chooses a more extensive treatment, the standard treatment is a radical hysterectomy with pelvic lymphadenectomy [25]. An intraoperative view of this procedure is presented in Figure 3.

A study by Ungar et al. emphasized the overwhelming importance of lymphadenectomy, as it is well known that, most frequently, recurrences originate from the regional lymph nodes [43]. Thus, in a group of 492 patients, they concluded that the complete excision of the pelvic lymph node tissue leads to a higher survival rate for operable cervical cancers, even in the absence of any adjuvant treatment, with results comparable to that of the less invasive techniques with adjuvant treatment [43]. However, in stage IB1, the sentinel node technique can also be used to minimize the postoperative risks of lymphadenectomy, but the tumor must be below 2 cm in size, the stromal invasion must be minimal (below 1 cm), and there must be no lymphovascular invasion [44].

A novelty in the approach of these stages is represented by image-guided interstitial irradiation for “bulky” tumors, with superior results to intracavitary brachytherapy and also associated with low toxicity, according to the EMBRACE (Image guided External radiochemotherapy and MRI-based BRAchytherapy in locally advanced CErvical cancer) and retroEMBRACE studies [35]. This technique can be used up to stage IIIB [45]. Additionally, a randomized clinical trial led by Gupta indicated that radiotherapy, even in the standard form, in combination with cisplatin, leads to a higher disease-free survival than the combination of neoadjuvant chemotherapy followed by radical hysterectomy and then adjuvant radiotherapy [46]. However, Landoni et al. state that, at this stage, no survival evidence overwhelmingly tilts the balance towards one treatment or another, concluding that the approach must be based on the particularities of the patient [47].

If the patient refuses surgery or her general condition contraindicates surgery, then treatment is limited to radiotherapy with or without chemotherapy [25].

The prognosis remains good, with a 5-year survival of 83.6%, the best in stage IB1—91.6%. If the patient is infected with HPV strain 18, she needs a more careful follow-up and rigorous periodic controls, more than in the case of other strains [48]. It should be noted that the use of image-guided brachytherapy, according to the EMBRACE and retroEMBRACE studies, led to a significant increase in survival, up to 96% at 3 years [49].

Regarding stage IIA1, an important aspect is highlighted by the study published by Chen et al. of 724 patients in which they compared overall and disease-free survival at 5 years in two groups: a group in which radical abdominal hysterectomy was performed and a group in which they performed only radiochemotherapy, the result being clearly in favor of hysterectomy [50]. The 5-year survival prognosis is 70.3% [34]. In extremely well-selected cases, in young patients, and if the preservation of fertility is attempted, it can be opted for trachelectomy [50].

### 4.4. Stages IB3 and IIA2

The tumor invades the upper 2/3 of the vagina, but without invasion of the parameters, and exceeds 4 cm in size [35].

The NCCN guidelines establish the following treatment alternatives: firstly, external radiotherapy associated with concurrent chemotherapy with platinum salts followed by brachytherapy, this approach being accepted in a uniform consensus. Another alternative treatment is radical hysterectomy with pelvic lymphadenectomy accompanied or not by para-aortic lymphadenectomy. In stages IB3 and IIA2, surgical treatment is recommended only as an isolated option on highly selected cases and only after the general consensus of a multidisciplinary committee [25]. The last alternative is external pelvic radiotherapy associated with concurrent chemotherapy with platinum salts and brachytherapy followed by complementary hysterectomy for certain selected cases. Survival in stage IB3 is 76.1% [34].

In stage IIA2, which is a locally advanced stage and the tumor exceeds 4 cm—the so-called “bulky” tumor, the standard of treatment is a combination of radio- and chemotherapy, and the survival seems comparable to other more radical methods of treatment [50,51]. However, according to NCCN guidelines, some experts express the opinion that at this stage, a pelvic lymph node dissection can be performed, and if there is no lymph node invasion, then a radical hysterectomy can be executed. If there is lymph node invasion, then the hysterectomy should be abandoned, and patients should undergo chemoradiation [25].

The 5-year survival prognosis in stage IIA2 is 65.3% [34].

In the guide of the Oncological Institute “Prof. Dr. Al. Trestioreanu” Bucharest, these stages are treated similarly to the other locally advanced stages (stage IB3-IV) [52].

### 4.5. Stage IIB

At this stage, most of the therapeutic guidelines—European Society of Gynecological Oncology (ESGO), European Society for Radiotherapy & Oncology (ESTRO), and NCCN recommend concurrent chemoradiotherapy (external irradiation 45–50 Gy + intracavitary brachytherapy up to 85–90 Gy simultaneously with Cisplatin-based chemotherapy), with the possibility of additional external irradiation with 5–10 Gy, in the case of parametrial invasion, as well as irradiation of the paraaortic lymph nodes [28].

Guide of the Oncological Institute “Prof. Dr. Al. Trestioreanu” Bucharest follows the European treatment guidelines but keeps an alternative variant for cases in which it is considered that the optimal radiation dose for lymph node involvement cannot be reached, this dose varying between 54 and 63 Gy. In these cases, the first step recommended is concurrent radiochemotherapy, similar to that proposed by the guidelines—external irradiation 45–50 Gy + intracavitary brachytherapy 15 Gy + Cisplatin-based chemotherapy [52]. However, this step is followed after an interval of 6–8 weeks by surgical intervention, which consists of different approaches depending on the extent of the lesions—e.g., radical hysterectomy + pelvic lymphadenectomy with or without para-aortic lymph node sampling or para-aortic lymphadenectomy for curative purposes, if there is an invasion of the paraaortic nodes [52].

This approach is based on the results of a study that showed that out of a total of 461 patients with locally advanced cervical cancer from the Oncological Institute “Prof. Dr. Al. Trestioreanu” Bucharest who was initially treated with chemoradiotherapy and then adjuvant surgical intervention was performed, in 254 patients, residual tumors were found at the histopathological examination. The response to treatment varied according to the histological subtype of cervical cancer and was 50.6% in cases of squamous cell carcinoma and 77.6% in cases of adenocarcinoma or adenosquamous carcinoma [53].

A series of other studies have shown that in stage IIB when the tumor invades the parameters but without touching the walls of the pelvis, the standard of treatment is combined chemoradiotherapy [35]. However, they specify that in some countries in Europe, the combination of neoadjuvant chemoradiotherapy is also recommended to be followed by radical hysterectomy and pelvic lymphadenectomy [54,55]. Probably the differences in approach are due to the perception related to radical hysterectomy, as well as from the attempt to minimize the toxicity of radiochemotherapy treatment [54]. This approach is also mentioned in the NCCN guidelines, where it is stated that in locally advanced stage disease, including FIGO stage IIB, in the United States, patients are treated with definitive chemoradiotherapy, but there are some countries, in which selected cases, stage IIB patients, can be treated with initial radical hysterectomy or neoadjuvant chemotherapy followed by radical hysterectomy [25].

On the other hand, it is worth mentioning that another study has shown higher morbidity, especially urinary, after postradiotherapy surgery, compared to definitive concurrent radiochemotherapy, with comparable overall survival [56]. Probably, the best approach is individual-based, after a complete evaluation of the risks and benefits and also an independent evaluation of the experience of the surgical team.

The 5-year survival prognosis in stage IIB is 63.9% [34]. However, using image-guided brachytherapy, the EMBRACE and retroEMBRACE trials showed survivals of up to 89%, although these were reported at 3 years, not at 5 [49].

### 4.6. Stage III

It is subdivided into three stages: IIIA—the tumor invades the lower third of the vagina, without invasion of the pelvic wall; IIIB—the tumor invades the pelvic wall, with or without hydronephrosis and concomitant renal damage; and stage IIIC—the tumor invades the pelvic and/or para-aortic lymph nodes, regardless of its extension or dimensions, according to FIGO 2018 staging [35].

Concomitant cisplatin-based chemoradiotherapy is the standard of therapy, with surgery being associated with higher morbidity, especially when combined with adjuvant radiotherapy [25,57,58]. Additionally, comparing survival in patients who received concurrent chemoradiotherapy with those who received radiotherapy alone at the same dose, it was found that there was an improvement in prognosis at 5 years in the combined chemoradiotherapy category [58]. It should be noted that the frequency of hematological and gastroenterological adverse reactions was higher in platinum salt-based therapies than in those in which another chemotherapy was used [59].

As a surgical treatment, there is a case published by J.P. Micha and J.V. Brown in 1998 in which a stage IIIB cervical cancer patient with ovarian, round ligament, and fallopian tube metastases who had an indication for total pelvic exenteration had an 8-year disease-free survival in the absence of any chemo- or radiotherapy treatment [60]. This method is used for recurrent cancer, despite the high morbidity and the need for careful patient selection [61]. The prognosis of patients at 5 years is 45% [34]. A negative prognostic factor is the invasion of the para-aortic nodes, with survival decreasing to 40% at 5 years [62]. However, even at this stage, imaging-guided brachytherapy can increase the reported 3-year survival to 73% [34,49].

Palliative care also plays a key role in the management of stage III cervical cancer, especially in cases where the cancer is advanced or not responsive to curative treatment.

Palliative care can be provided alongside curative treatments or as the primary focus of care, depending on the individual’s needs and treatment goals. It is an essential component of comprehensive cancer care and helps ensure that patients with stage III cervical cancer receive holistic support, symptom management, and improved quality of life [63].

### 4.7. Stage IV

The tumor extends to adjacent organs, such as the urinary bladder and rectum, or distant metastases are identified [35]. The prognosis is reserved, with 5-year survival in stage IVA, in which the tumor invades only neighboring organs up to 24%, and in stage IVB, with distant metastases, only 14.7% [34].

The independent prognostic factors reported in the literature are the performance status, the location of the metastases, and the local invasion of the disease. Because there is a large individual variation in the evolution of the disease at this stage, it has long been considered difficult to establish a standard approach.

The most commonly used treatment is, as in the case of stage III, chemotherapy based on platinum salts and radiotherapy [64]. Subsequently, if a satisfactory post-chemotherapy response is obtained, tumor excision surgery and metastasectomy can be considered depending on the biological status of the patient and the number and resectability of metastases [64].

Figure 4 presents a patient with a bulky ilio-obturatory lymph node metastasis.

During treatment, immunotherapeutic agents such as bevacizumab and pembrolizumab can be administered [65].

For recurrent or persistent centrally located pelvic disease after radiotherapy, pelvic exenteration (pelvectomy) can be considered [66]. Preoperative evaluation for this procedure must exclude distant metastases. Additionally, the intraoperative exploration must assess the possibility of complete resection, which means that the resection margins do not contain any neoplastic tissue at the histopathological examination [66].

Depending on the tumor location, resection may include anterior exenteration, posterior exenteration, or total exenteration [66]. In cases where the location of the tumor allows the achievement of satisfactory oncological safety margins without resection of the pelvic floor and the anal sphincter muscle, they can be preserved (supralevator exenteration) [66].

It must be highlighted that pelvectomies are extremely complex procedures and must be performed in centers with a high level of experience and where multidisciplinary surgical teams can be assembled [61].

Primary pelvectomy (without prior pelvic irradiation) can also be performed, but it is limited to the rare cases where irradiation is contraindicated or to patients who have previously undergone pelvic irradiation for another cancer and then developed a metachronous, locally advanced cervical tumor [61].

Palliative care is very important in the management of stage four cervical cancer; it aims to improve the overall quality of life, providing comfort, relief from symptoms, and emotional support throughout the disease course. It focuses on a multidisciplinary approach involving healthcare professionals from various specialties to address the complex needs of patients and their families [63].

## 5. Future Improvement Strategies in Cervical Cancer Treatment

Despite advances in screening, prevention, and treatment, cervical cancer remains a significant global health issue, particularly in middle-income countries. Some areas where improvement is needed include increased access to affordable and accurate screening methods, such as HPV testing and cytology (Pap tests) [67].

Improving HPV vaccination coverage is also vital for primary prevention. Expanding vaccination programs and addressing barriers, such as cost, awareness, and vaccine availability, can help reduce the incidence of cervical cancer in the long term [67].

Strengthening healthcare infrastructure and resources in middle-income countries is essential to ensure timely and quality treatment for cervical cancer. This includes improving access to surgery, radiation therapy, chemotherapy, and supportive care services.

Building and enhancing the skills of healthcare professionals in the diagnosis and management of cervical cancer through training programs, continuing education initiatives, and knowledge-sharing platforms can help healthcare providers stay updated with the latest advancements in cervical cancer treatment.

Additionally, encouraging research efforts focused on developing novel therapies, targeted treatments, and more effective treatment protocols for cervical cancer can contribute to better outcomes.

Targeted therapy has emerged as an important approach in the treatment of various cancers [68]. However, its use for cervical cancer is currently limited compared to other types of cancer. Nonetheless, ongoing research and clinical trials are exploring the potential of targeted therapies in cervical cancer treatment. For example, bevacizumab has been investigated in combination with chemotherapy in advanced cervical cancer and has shown improvement in overall survival [69]. Pembrolizumab has shown activity in advanced or recurrent cervical cancer and has been approved for certain patients with advanced disease [70]. Niraparib and rucaparib have shown promise in clinical trials for the treatment of recurrent or metastatic cervical cancer, particularly in patients with certain genetic mutations, such as BRCA1/2 [71].

Clinical trials continue to investigate the efficacy and safety of targeted therapies, both as monotherapies and in combination with other treatments.

Providing comprehensive support services, including psychosocial support, counseling, and patient education, can improve patient well-being and treatment adherence.

Efforts to address these areas can help bridge the gaps in cervical cancer treatment and ultimately reduce the burden of the disease. Collaboration between governments, healthcare organizations, researchers, advocacy groups, and international partners is crucial in driving these improvements and making cervical cancer care more accessible, effective, and equitable.

## 6. Conclusions

Cervical cancer is an important cause of mortality in fertile women. Screening programs and vaccination against oncogenic strains of HPV are major means of prevention against cervical cancer. Any positive screening test should be confirmed by microscopy and imaging investigations to determine the stage of the disease.

In the early stages (IA1, IA2), which are considered micro-invasive, cervical conization or trachelectomy is recommended if the patient wishes to preserve fertility; otherwise, a simple hysterectomy is recommended. If the patient refuses surgical intervention, she can opt for neoadjuvant radiotherapy and periodic reassessment.

In the invasive stages—IB1 to IIA1, the standard treatment is radical hysterectomy with pelvic lymphadenectomy followed by chemoradiotherapy. In advanced stages such as IIB to IVA, chemoradiotherapy remains the standard of treatment. Surgical intervention can be considered, but only in selected cases. However, this approach does not always provide a real benefit to patients’ survival if complete resection of the tumor and metastases is not achieved, and may also lead to delays in the administration of treatments and to the exposure of the patients, who sometimes have precarious physiologic status, to anesthetic-surgical risks.

Regarding our country, due to the lack of effective prevention programs and the reluctance of patients toward vaccination, the stages of presentation in the health system are sometimes advanced and, unfortunately, often accompanied by complications such as profuse bleeding with secondary anemia. In these situations, surgical intervention may be indicated for hemostatic and palliative purposes.

## Figures and Tables

**Figure 1 medicina-59-01229-f001:**
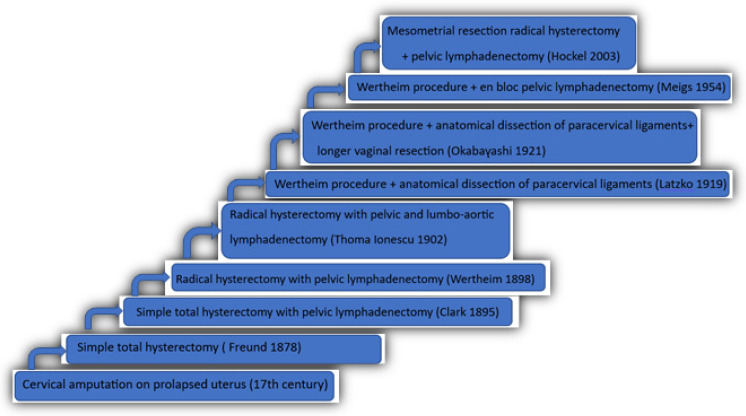
Evolution of surgery in genital cancer [14,15,16,17,18,19,20,21].

**Figure 2 medicina-59-01229-f002:**
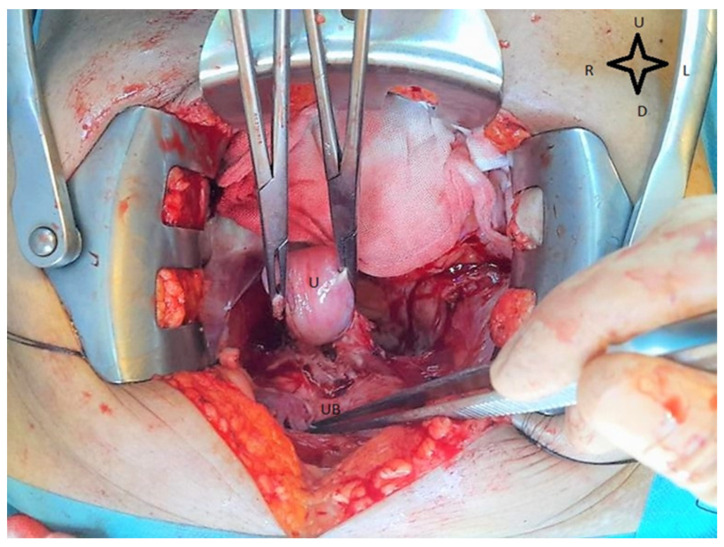
Intraoperative view during radical hysterectomy. Ovaries and fallopian tubes are removed, and uterus is tractioned for easier separation of cervix and posterior bladder wall. UB—urinary bladder, U—uterus, U—up, D—down, R—right, L—left.

**Figure 3 medicina-59-01229-f003:**
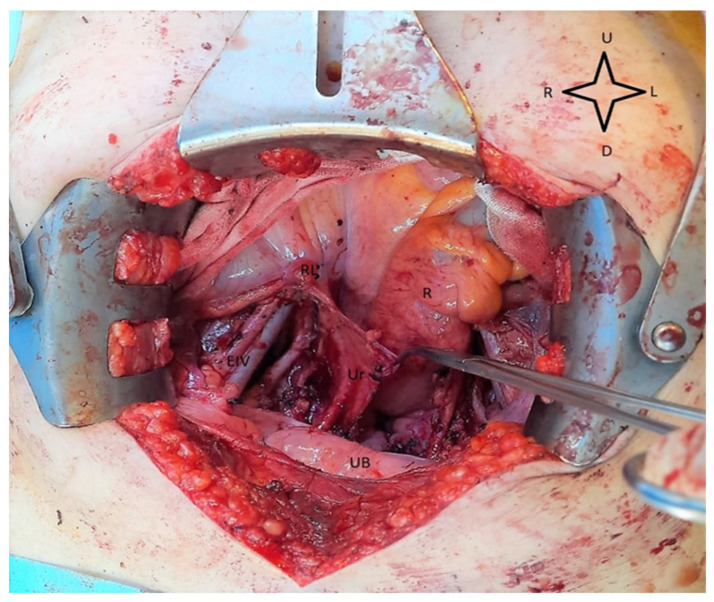
Intraoperative view during ilio-obturatory lymphadenectomy. The external iliac vessels and ureter are visualized, and the surrounding fat-containing lymph nodes are removed. RL—Round ligament ligated, R—rectum, Ur—right ureter, EIV—External iliac vessels, UB—urinary bladder, U—up, D—down, R—right, L—left.

**Figure 4 medicina-59-01229-f004:**
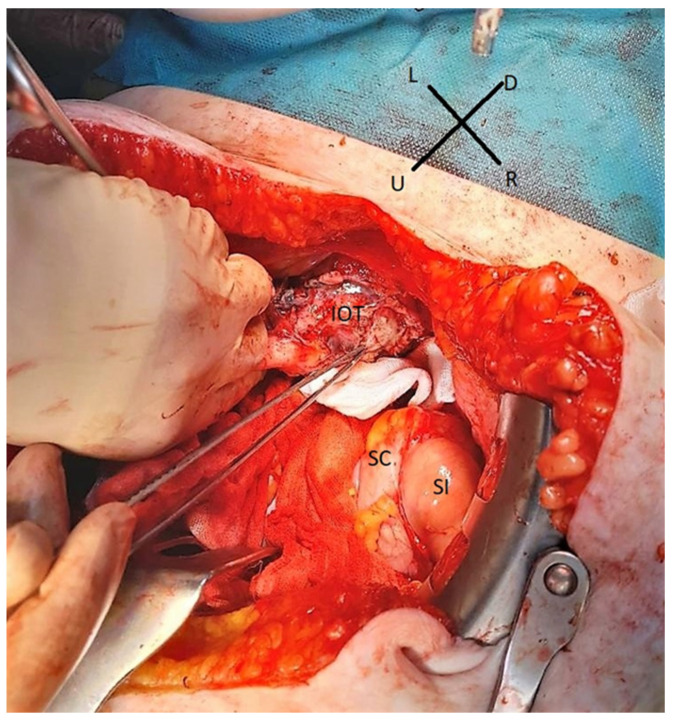
Intraoperative view of a patient during pelvic exenteration with bulky ilio-obturatory metastasis. SC—sigmoid colon. SI—small intestine. IOT—ilio-obturatory tumor. U—up, D—down, L—left, R—right.

**Table 1 medicina-59-01229-t001:** FIGO (2018) classification in cervical cancer.

FIGO	Description
I	Invasive carcinoma confined to the cervix.
IA	Invasive carcinoma can only be diagnosed microscopically with a maximum depth of invasion ≤5 mm. It affects the lymphovascular space.
IA1	Stromal invasion ≤ 3 mm in depth.
IA2	Stromal invasion ˃ 3 and ≤5 mm in depth.
IB	Invasive carcinoma, with an invasion > 5 mm, with the lesion limited to the cervix.
IB1	Invasive carcinoma with stromal invasion ˃ 5 mm and size ≤ 2 cm.
IB2	Invasive carcinoma ˃ 2 cm and ≤4 cm in the largest dimension.
IB3	Invasive carcinoma ˃ 4 cm in the largest dimension
II	Carcinoma invades beyond the uterus but does not extend to the lower third of the vagina or to the pelvic wall.
IIA	Invasion is limited to the upper two-thirds of the vagina but without parametrial invasion.
IIA1	Invasive carcinoma ≤ 4 cm in the largest dimension.
IIA2	Invasive carcinoma ˃ 4 cm in the largest dimension
IIB	Parametrial invasion is present, but the tumor does not reach the pelvic wall.
III	Carcinoma invasion to the lower third of the vagina, and/or extension to the pelvic wall and/or causes hydronephrosis or non-functioning kidneys.
IIIA	Carcinoma invasion up to the lower third of the vagina, without extension to the pelvic wall.
IIIB	Extension to the pelvic wall and/or hydronephrosis or non-functioning kidney.
IIIC	Invasion of the pelvic and/or the para-aortic lymph nodes, regardless of tumor size and extent.
IIIC1	Metastases in pelvic lymph nodes only.
IIIC2	Paraaortic lymph node metastases.
IV	Carcinoma that invades beyond the pelvis or has invaded the mucosa of the bladder or rectum (invasion confirmed by biopsy).
IVA	Metastasis in adjacent pelvic viscera.
IVB	Metastasis in other distant organs.

## Data Availability

Not applicable.

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
