# Peer review of "Current Therapeutic Approaches in Cervical Cancer Based on the Stage of the Disease: Is There Room for Improvement?"

_medicina, 2023, doi:10.3390/medicina59071229_

Round 1

Reviewer 1 Report

This work is a review of the treatment of cervical cancer, with a scheme similar to the international guidelines.

From this point of view, the ESGO/ESTRO/ESP Guidelines for the management of patients with cervical cancer has been published recently and the authors do not reference it (Cibula D, Raspollini MR, Planchamp F, et al. Int J Gynecol Cancer 2023;33:649–666.). Due to this fact, some of the assertions made by the authors do not strictly follow the current recommendations of the European Society of Gynecological Oncology as for example: 

-       Restrict the use of the sentinel node technique to clinical trials.

-       Do not emphasize that the surgical treatment of stages IB3 and IIA2 should be an isolated option in highly selected cases and always under the consensus of the local multidisciplinary committee.

-       Do not comment on the LACC study, one of the studies that has had the greatest impact in the surgical management of cervical carcinoma in its initial stages in recent times.

-       The use of neoadjuvant chemotherapy for the preservation of fertility in stage IB2 tumors should be considered as a possibility, but not with the degree of assertion given by the authors, because there is still not enough evidence to establish it as a standard treatment. 

Author Response

First of all, thank you for your effort and patience to read this review. Indeed, there were some gaps in our article and we hope that after the changes we made, it meets your legitimate requests.

1.″ESGO/ESTRO/ESP Guidelines for the management of patients with cervical cancer has been published recently and the authors do not reference it″

We are sorry to have overlooked the publication of these guidelines. After your suggestion, we thoroughly researched them and modified the article accordingly.

  1. ″Restrict the use of the sentinel node technique to clinical trials. ″

Indeed, now the sentinel node technique is used more frequently. We corrected this mistake.

  1. ″Do not emphasize that the surgical treatment of stages IB3 and IIA2 should be an isolated option in highly selected cases″

Thank you very much for pointing this out! We also changed this and hopefully, the information is more appropriate now.

  1. ″Did not comment on the LACC study. ″

We also changed this aspect. Now the study is quoted and commented on throughout the article. Thank you!

  1. ″The use of neoadjuvant chemotherapy for the preservation of fertility in stage IB2 tumors should be considered as a possibility, but not with the degree of assertion given by the authors″

We did not want it to sound like we were too confident in recommending chemotherapy but, because you mentioned, we emphasized that it is only an option, not a standard practice.

Thank you once again for your kind advice and suggestions!

Reviewer 2 Report

According to the abstract, the aim of the authors was to discuss  the diagnosis and treatment options in cervical cancer depending on the histological type, FIGO staging, and patient performance index, taking into account the hospital resources available in middle-income countries , however, in the article, the authors limited themselves just to discussing the methods of treatment used in various stages of cervical cancer

The authors did not answer the question asked in the title (is there room for improvement?).  When asking this question, did the authors mean changes in middle-income countries or maybe advances in immunotherapy, targeted therapy or genetic approaches to treating cervical cancer ?

Some minor comments below:

Part 4:

Stage IA1:                                                                                                      line 140: according to the FIGO classification of 2018, the IA1 stage refers to the depth of stromal infiltration up to 3mm. The use of a transverse dimension has been removed from FIGO classification

Stage IA2:                                                                                                           line 172: according to the FIGO classification of 2018, the IA2 stage refers to the depth of stromal infiltration up to 5mm. The transverse dimension is not taken into account (see reference 35)

Stages IB1, IB2, IIA1:                                                                                         lines 207- 208: the authors write:..If the tumor is larger than 2cm but less than 4cm  neoadjuvant  therapy can be chosen „… and refer (25) to the recommendations of the NCCN. The NCCN recommendations mention neoadjuvant chemotherapy as an option for the management of small cell neuroendocrine carcinoma when the tumor size is greater than 4 cm followed by interval hysterectomy or chemoradiationn + brachytherapy

Stages IB3 and IIA2:                                                                                         lines 283-284: the authors write:…” If the lymphatic invasion is detected, hysterectomy and lymphadenectomy are completed with concurrent radio-chemotherapy”… and refer (25) to the recommendations of the NCCN. The NCCN recommendations mention that some panel members express the opinion that a pelvic lymph node dissection should be performed first and if negative, then the radical hysterectomy should be performed. If the lymph nodes are positive, then the hysterectomy should be abandoned; these patients should undergo chemoradiation. I am not sure of the intention of the authors in the mentioned sentence in lines 283-284: did they mean to refrain from continuing the surgery after confirming lymph node involvement ?

Reference number 50 was unfortunately not available to me

Author Response

Thank you very much for your effort in evaluating this review and also for pointing out some aspects that we have overlooked. We hope that the changes made meet your demands.

  1. ″According to the abstract, the aim of the authors was to discuss the diagnosis and treatment options in cervical cancer depending on the histological type, FIGO staging, and patient`s performance index. However, in the article, the authors limited themselves just to discussing the methods of treatment used in various stages of cervical cancer. The authors did not answer the question asked in the title (is there room for improvement?)″

Thank you for your advice! Indeed, the answer to the question posed in the title was not specifically expressed. To fix this issue we added a separate paragraph in which we discussed what further treatment options can be to lower the mortality associated with advanced stages of cervical cancer. Thank you again!

Regarding the fact that we seemed to discuss only the treatment methods in different stages of the disease, indeed, we focused on them, but we also mentioned the connection between the histological type and the surgical technique used and also what treatment options are available if the patient`s performance index is low in different stages, even if, indeed, we did not explicitly use this expression, but, eventually general condition of the patient. However, to be sure that all the aspects you suggested are met, we also added more information about the influence of palliative care in advanced stages, which is of the utmost importance when the patient`s performance index is low.

  1. line 140: stage IA1 refers to the depth of stromal infiltration up to 3mm.

It was a translation mistake, we modified it. Thank you!

  1. line 172, stage IA2 refers to the depth of stromal infiltration up to 5 mm.

The same mistake as above, we corrected it. Thank you again!

  1. Stages IB1, IB2, IIA1: lines 207-208: the authors write: "If the tumor is larger than 2cm but less than 4cm neoadjuvant therapy can be chosen „… and refer (25) to the recommendations of the NCCN. The NCCN recommendations mention neoadjuvant chemotherapy as an option for the management of small cell neuroendocrine carcinoma when the tumor size is greater than 4 cm followed by interval hysterectomy or chemoradiation + brachytherapy”.

Thank you for your observation! We modified the paragraph and mentioned neoadjuvant therapy as an option, not as a standard treatment, due to lack of evidence of its efficiency.

  1. Stages IB3 and IIA2: lines 283-284: the author writes:” If lymphatic invasion is detected, hysterectomy and lymphadenectomy are completed with concomitant radio-chemotherapy” …and refer (25) to the recommendations of the NCCN. The NCCN recommendations mention that some panel members express the opinion that a pelvic lymph node dissection should be performed first and if negative, then the radical hysterectomy should be performed. If the lymph nodes are positive, then the hysterectomy should be abandoned; these patients should undergo chemoradiation. I am not sure of the intention of the authors in the mentioned sentence in lines 283-284: did they mean to refrain from continuing the surgery after confirming lymph node involvement ?

Yes, we meant to refrain from continuing the surgery if there are positive lymph nodes. However, indeed, the phrase was ambiguous and, therefore, we reformulated it according to your advice to be clear. Thank you for pointing this out!

  1. Reference number 50 was unfortunately not available to me.

This was the reference to the guidelines of the Ministry of Health in Romania regarding the treatment for cervical cancer from this moment. After revision, it became the reference nr. 52. We updated it with the correct web page. We hope that now you can visualize it.

Thank you once again for your kind advice and suggestions!

Round 2

Reviewer 2 Report

I would like to thank the authors for the opportunity to read the revised version of the article. The addition of a section entitled "Future improvement strategies in cervical cancer treatment", in which the authors succinctly summarized the most important "repair" points that are valid with different weight at different latitudes, is the proverbial dot over i. I have no more comments. In my opinion, the article in its current version deserves publication